# Preventing Dementia—A Cross-Sectional Study of Outpatients in a Tertiary Internal Medicine Department

**DOI:** 10.3390/jpm13121630

**Published:** 2023-11-22

**Authors:** Ioana-Alexandra Sandu, Ramona Ștefăniu, Teodora Alexa-Stratulat, Adina-Carmen Ilie, Sabinne-Marie Albișteanu, Ana-Maria Turcu, Călina-Anda Sandu, Anisia-Iuliana Alexa, Anca-Iuliana Pîslaru, Gabriela Grigoraș, Cristinel Ștefănescu, Ioana-Dana Alexa

**Affiliations:** 1Department of Medical Specialties II, University of Medicine and Pharmacy “Grigore T Popa”, 700115 Iasi, Romania; ioana-alexandra.sandu@umfiasi.ro (I.-A.S.); teodora.alexa-stratulat@umfiasi.ro (T.A.-S.); adina.ilie@umfiasi.ro (A.-C.I.); sabinne-marie.taranu@umfiasi.ro (S.-M.A.); ana-maria_turcu@umfiasi.ro (A.-M.T.); calina-anda_c_sandu@d.umfiasi.ro (C.-A.S.); anisia-iuliana.alexa@umfiasi.ro (A.-I.A.); anca.morosanu2@umfiasi.ro (A.-I.P.); grigoras_gabriela@d.umfiasi.ro (G.G.); cri.stefanescu@umfiasi.ro (C.Ș.); ioana.alexa@umfiasi.ro (I.-D.A.); 2Geriatrics and Internal Medicine Department, “C. I. Parhon” Hospital, 700503 Iasi, Romania; 3Ophtalmology Department, “St. Spiridon” Hospital, 700111 Iasi, Romania; 4Acute Psychiatry Department, “Socola” Institute of Psychiatry, 700282 Iasi, Romania

**Keywords:** screening, dementia, mCAIDE, outpatient clinic

## Abstract

Dementia is a significant health problem worldwide, being the seventh leading cause of death (2,382,000 deaths worldwide in 2016). Recent data suggest there are several modifiable risk factors that, if addressed, can decrease dementia risk. Several national dementia screening programs exist; however, limited-income countries do not have the means to implement such measures. We performed a prospective cross-sectional study in an outpatient department to identify individuals at risk for dementia. Patients with no known cognitive dysfunction seeking a medical consult were screened for dementia risk by means of the cardiovascular risk factors, ageing, and dementia (CAIDE) and modified CAIDE tests. Additionally, we collected demographic and clinical data and assessed each participant for depression, mental state, and ability to perform daily activities. Of the 169 patients enrolled, 63.3% were identified as being in the intermediate-risk or high-risk group, scoring more than seven points on the mCAIDE test. Over 40% of the elderly individuals in the study were assessed as “somewhat depressed” or “depressed” on the geriatric depression scale. Almost 10% of the study population was diagnosed de novo with cognitive dysfunction. In conclusion, using a simple questionnaire such as the mCAIDE in a predefined high-risk population is easy and does not represent a major financial burden. At-risk individuals can subsequently benefit from personalized interventions that are more likely to be successful. Limited-resource countries can implement such screening tools in outpatient clinics.

## 1. Introduction

Dementia is often incorrectly referred to as “senility” or “senile dementia,” reflecting the widespread but incorrect belief that severe mental decline is a normal part of aging. Dementia is not a single disease; it is an umbrella term covering a wide range of distinct medical conditions that progressively affect memory, thinking, and social abilities [1]. Worldwide, dementia is considered the seventh leading cause of death and among the top ten causes of disability in the elderly [2]. A systematic analysis for the Global Burden of Disease Study published in 2019 estimated that Alzheimer’s disease and other dementias are among the top four contributors to global neurological disability-adjusted life-years (DALYs—the sum of years of life lost and years lived with disability). The age-standardized prevalence, death, and DALYs rates were 712, 41 and 471 per 100,000 individuals, respectively [3]. Additionally, individuals with dementia have a significant psychological and social impact on both their families and their formal and informal caregivers. Another 2019 analysis estimated that there are approximately 55.2 million people living with dementia, and the annual global cost of care is USD 1313.4 billion (USD 23,796 per person), a significant increase from the 2010 (USD 604 billion) and 2015 (USD 818 billion) estimates [4].

Most dementias are associated with structural and/or functional brain changes, including Alzheimer’s disease (AD), which accounts for 60–80% of all dementia cases [5,6]. Over time, these changes trigger a decline in cognitive skills severe enough to affect daily life and independent function, impairing behavior, feelings, and relationships. Several drugs, such as galantamine, rivastigmine, donepazil, and memantine can improve cognitive function in individuals diagnosed with AD [7]. Similarly, intravenous cerebrolysin improves cognition and general function in patients with vascular dementia [8]. Nonetheless, these drugs address disease symptoms and not pathogenesis, and thus dementias remain incurable illnesses despite numerous clinical trials of promising drugs and interventions in the past decades [9,10].

Given the disappointing results for curing dementias, there has been a recent shift in approach, with more focus placed on identifying individuals at risk for developing dementia. Data reported at the 2019 Alzheimer’s Association International Conference showed that around 40% of dementia cases could be the result of modifiable risk factors. Some dementia risk factors, notably cardio-vascular, are well known and have been investigated across multiple studies. Hypertension is linked to several types of dementia due to subsequent vascular changes and small-vessel ischemia [11]. Chronic high blood pressure is also associated with changes in cerebral vasculature, impairing Aβ clearance and increasing Aβ deposits, thus increasing AD risk [12]. However, in recent years, more dementia risk factors have been identified and current literature suggests there are in fact twelve potentially modifiable risk factors: high blood pressure, smoking, diabetes, excessive alcohol consumption, obesity, lack of physical activity, hearing impairment, traumatic brain injury, depression, low social contact, air pollution, and less education [13].

Intervention strategies focused on modifiable risk factors for the disease are becoming a realistic and relevant therapeutic strategy for Alzheimer’s prevention [14]. Epidemiological evidence of AD risk factors is contributing to and encouraging the development of primary prevention initiatives; identifying individuals at risk of developing the disease might be the key to the success of future intervention studies. Current estimates show that the age-specific incidence of dementia is following a downward trend, possibly due to better management of both diabetes and hypertension and an increase in literacy [15], although changes in other of the afore-mentioned risk factors could also play a role.

Early identification of cognitive dysfunction can help to plan for pharmacological and non-pharmacological interventions that increase quality of life and prolong the time to severe cognitive dysfunction [16], while screening for potentially modifiable dementia risk factors can significantly decrease the number of individuals affected by dementia. Although several economic analyses favoring this approach have been published [17,18], many countries still cannot implement a national screening program due to a lack of funds and personnel. As such, easy-to-implement alternatives used in a selected high-risk population can partly address this important issue. Several of the twelve afore-mentioned dementia risk factors can be a reason for an individual to seek a medical consultation in an outpatient clinic. As such, adding a simple screening test for assessing dementia risk in the initial outpatient evaluation would incur minimal additional work and could identify at-risk individuals, thus serving as a low-cost screening tool [19].

Currently, several short and reliable tests are available in clinical practice for estimating future dementia risk [20]. Among them, the cardiovascular risk factors, ageing, and dementia (CAIDE) tool is widely used due to its ease and large-scale validation. It was developed over 15 years ago to assess the risk of dementia in midlife adults [21,22]. Since then, the demographic shift has become even more apparent, with a steady increase in the average life expectancy (from 70.1 to 73.16), which suggests that the elderly should also be screened for dementia since they are expected to live longer. As such, the modified CAIDE (mCAIDE) tool was developed and validated in 2021 [23] and has since been used in clinical practice due to its increased ease and ability to correctly predict dementia risk in both midlife and elderly adults [24,25]. 

The main objective of this study was to assess dementia risk by means of the modified CAIDE (mCAIDE) score in the internal medicine department of a tertiary hospital in NE Romania and compare it to available data from the general population. Secondary objectives included identifying potential correlations between the mCAIDE score and different social and clinical patient data and comparing the CAIDE and the mCAIDE in non-elderly individuals. To the author’s knowledge, this is the first study to report dementia risk using the mCAIDE score in Eastern Europe.

## 2. Materials and Methods

### 2.1. Study Design

Consecutive individuals during their first non-emergency visit to the Internal Medicine/Geriatrics outpatient department of Parhon Hospital were prospectively included in this cross-sectional study over a period of four months (November 2021–February 2022). The main exclusion criteria were the patient’s inability or lack of willingness to participate in the study or sign the informed consent form, age < 18 years, a known diagnosis of dementia/cognitive impairment, and hospital admission due to an acute medical condition (emergency hospitalization). A prior diagnosis of a cardio-vascular or metabolic condition did not preclude study enrollment. 

The institutional ethics board approved the study. All procedures performed in this study followed the Declaration of Helsinki. The study protocol was approved by the Ethics Committee of the “C.I. Parhon” hospital, Iași (Approval number 3340/04.05.2021). Informed consent was obtained from all subjects involved in the study.

Patients provided information on sociodemographic characteristics (educational level, smoking status) and personal medical history (a history of diabetes, hypertension, or abnormal cholesterol levels) as part of the standard anamnesis at admission (interview). Subsequently, they underwent cognitive screening by means of the Mini-Mental State Evaluation (MMSE). A clinical examination was performed as per local standards, additionally including measurements of the brachial and calf circumference and the mini-physical performance test (mPPT) [26]. The mPPT is an easy-to-use 4-item test of physical mobility that is included in the mCAIDE scoring system. It is derived from the more extensive physical performance test (PPT) that has been validated in several populations, including individuals with dementia [27].

For each patient, assessments for dyslipidemia (total cholesterol, HDL-cholesterol, LDL-cholesterol, triglycerides, and uric acid) and diabetes (fasting blood glucose) were recorded. 

Nutritional status was assessed by means of several tests: calf and brachial circumference, body mass index (BMI), and Mini-Nutritional Assessment (MNA). The latter is a 12-item questionnaire that consists of an initial screening part and a subsequent assessment part that aims to identify malnourished individuals or individuals at risk for malnutrition by revealing different etiological factors or relevant symptoms such as acute stress, neuropsychological issues, mobility, food, fluid, and protein intake, ability to self-feed, and self-perception about health. It is primarily used in elderly individuals but is also used and validated in adult populations [28]. Depression screening was performed by means of the Hospital Anxiety and Depression Scale (HADS) for non-elderly individuals and the Geriatric Depression Scale (GDS) for elderly individuals. Additionally, individuals aged over 65 were also assessed by means of the Activities of Daily Living (ADL) scale and the Instrumental Activities of Daily Living (IADL) scale as per local protocol.

### 2.2. Dementia Screening

All enrolled patients were subsequently screened for dementia risk. The CAIDE score is a validated clinical tool developed for non-elderly individuals that takes into account seven parameters—age, gender, systolic blood pressure, physical activity, education, body mass index, and total cholesterol—and offers a final score (0 to 15) that is subsequently converted into a percent risk of developing dementia over the next 20 years [29]. For example, a score between 0 and 5 points equals a 1% risk of developing dementia, while a score between 12 and 15 equals a 16.4% risk of developing dementia. The mCAIDE tool, developed by Tolea and colleagues, has the same variables as the CAIDE tool (age, gender, systolic blood pressure, physical activity, education, body mass index, and total cholesterol), but scores and assesses some of them differently to reflect the inclusion of older adults and geographic educational characteristics [23]. As such, the mCAIDE reports educational level as <12 vs. 12–16 vs. >16 years, uses the miniPPT test for physical activity assessment (instead of active/inactive), and divides individuals according to age in ≥73 vs. 65–72 vs. <65 years categories. Additionally, it uses a yes/no system for scoring self-reported increased cholesterol levels instead of a formal value assessed by a laboratory to increase its ease to use in the clinical setting. The mCAIDE score ranges from 0 to 14 and individuals that have a score of 7 or above are considered to have intermediate or high-risk profiles and an increased risk for developing dementia in the future. In our study, adult individuals (<65 years) were screened via both CAIDE and mCAIDE tests in order to compare the two methods of assessment, whereas elderly individuals were screened by means of the mCAIDE test only.

### 2.3. Statistical Analysis

Statistical analysis was performed by means of SPSS software (version 20 for Windows; SPSS Inc., Chicago, IL, USA). Data are expressed as the mean ± SD, with minimum and maximum values, or as median, according to data type. Correlations were calculated using the Pearson, Kendall, or Spearman coefficients according to the data type. CAIDE and mCAIDE scores were calculated according to guidelines, and dementia risk was subsequently calculated in percent value. An unpaired Student *t* test was used to assess potential differences between groups. A *p* value < 0.05 was a priori considered statistically significant.

## 3. Results

We identified 169 patients who met the pre-specified inclusion criteria. The mean age was 64.43 ± 9.08 years. Reasons for outpatient clinic visits included general practitioner recommendations, routine check-ups of a pre-existing condition, or the appearance of a new sign or symptom, most often pain. The majority of patients were female, non-smokers with no known chronic illness (Table 1). MMSE screening revealed that most patients (92.3%) did not have cognitive dysfunction. However, test scores indicated mild, moderate, or severe cognitive impairment for nine (5.3%), three (1.8%), and one (0.6%) patients, respectively, even though exclusion criteria precluded individuals with known cognitive dysfunction from enrolling.

Nutritional evaluation concluded that most patients (60.9%) are obese due to values exceeding 30 kg/m^2^ as per the BMI calculator, and the remaining majority was considered either overweight or normal weight, with no underweight individuals detected. However, the MNA assessment identified 48 patients (28.4%) at risk for malnutrition and 11 patients (6.5%) with malnutrition. Of the 43 individuals with increased serum glucose at initial assessment, 21 (12.4%) were diagnosed with de novo Type 2 diabetes. A similar finding was reported for increased systolic blood pressure, with 26 (15.4%) new diagnoses of hypertension. 

The HADS screening assessment did not find individuals with borderline abnormal or abnormal results. However, in the elderly population segment, over 40% of those aged over 65 had abnormal results in the GDS assessment. Of the 108 elderly, 30 individuals (27.8%) were included in the “somewhat depressed” category, and an additional 15 (13.9%) were considered “depressed”.

Over 40% of study participants (42.6%) reported a history of increased cholesterol values and/or known dyslipidemia diagnosed by either their general practitioner or another physician during a prior hospital visit. A somewhat similar percentage of patients (39.6%) was found to have increased cholesterol levels after serum analysis. The comparison of the two measurements revealed non-significant statistical differences, despite the fact that some of the patients were receiving cholesterol-lowering drugs and some were diagnosed with dyslipidemia during this initial hospital visit. A detailed report on patient demographics and findings can be found in Table 1.

### 3.1. Geriatric-Specific Screening Tests

In the study group, 108 patients were over 65. These individuals were additionally screened for the ability to perform daily activities. Most elderly patients reported being able to perform all activities in the ADL assessment (75% of cases, 81 individuals), and only 8 patients (7.4%) were considered completely dependent. Similar findings were noted in the IADL evaluation, with normal scores in 69.4% of cases (75 patients) and 11.1% (12 patients) being considered dependent. 

### 3.2. Dementia Risk

All participants were evaluated by means of the mCAIDE assessment (Table 2). The average mCAIDE score for the entire group was 7.05 points, with a minimum of 1 point and a maximum of 12 points. Most participants were identified as being in the intermediate-risk or high-risk group, scoring more than 7 points on the test (107 participants, 63.3%). An additional 27.2% of study participants scored 5 or 6 points, making them potential intermediate-risk individuals in the future.

A higher dementia risk as identified by mCAIDE was correlated with abnormal results in the MMSE test and increased blood sugar. No correlations between specific lipid fractions (HDL-cholesterol, LDL-cholesterol, and triglycerides) and mCAIDE score were identified. In elderly patients, a correlation was identified between changes in ADL and a score larger than seven on the mCAIDE test.

In order to compare the two instruments, non-elderly individuals also underwent assessment using the traditional CAIDE tool. The mean CAIDE score of the 61 non-elderly individuals (mean age 54.4 years) included in the study was 7.98 ± 2.72 points, with a mean risk of developing dementia over the next 20 years of 4.61 ± 3.55%. Almost 30% of all adult individuals were assessed as having a likelihood of developing dementia that exceeded 7%, and one sixth of these were assigned a 16.4% likelihood. The results in the CAIDE score strongly correlated with mCAIDE results. A detailed analysis of the CAIDE and mCAIDE scores in the non-elderly population can be found in Table 3. 

## 4. Discussion

Life expectancy has doubled in the past two centuries, mostly due to increased knowledge of disease prevention and advances in treatment [30]. As such, several “classical” cut-offs, such as those used for separating midlife from old age, are being challenged, and overall, the population aging phenomenon has led to including individuals up to 60 in the midlife category [31]. Taking into account that the average life expectancy of a 65-year-old individual exceeds 20 years in some countries, prevention programs for debilitating conditions should be implemented in this age category as well. The mCAIDE tool is a simple screening method that can be used for predicting dementia risk, irrespective of age. Applying the mCAIDE to consecutive individuals visiting an outpatient internal medicine clinic revealed that over 60% of the study group have an intermediate or high risk for developing dementia. These results are in line with the available literature data, even though the inclusion criteria and baseline characteristics of study populations vary across studies. For example, an analysis of individuals included in the Northern Manhattan Study (NOMAS) (community-dwelling adults over 40 with no history of stroke) concluded that the median CAIDE score was 8, similar to the average of 7.98 points in our study [32]. The mCAIDE validation study used two population samples and reported an average mCAIDE score of 5.41 ± 2.89 points for community-dwelling adults and 7.92 ± 2.79 points for adults with or without cognitive impairment or dementia attending a South Florida academic dementia center. However, we found several differences between the demographic characteristics of our study group and those of the original mCAIDE validation group [23]. A significantly larger percent of individuals in our study were overweight, and the prevalence of obesity exceeded 60%. While reports of the World Obesity Federation Global Obesity Observatory estimate that obesity prevalence is approximately 10% in Romania [33], real-world studies report that approximately 40% of Romanians are obese [34]. Despite a significant percentage of overweight or obese individuals in our group, most likely due to sample selection, most (82.2%) had good results in the physical performance test. The majority of unfit patients stemmed from the adult population (28 out of 30), most likely because elderly individuals that are also unfit require more advanced medical help and are usually hospitalized or assessed in a non-outpatient setting. Notable differences were also present in education, with 64.7% of individuals reporting less than 12 years of education compared to an average of 13.7 years in the mCAIDE validation group. These differences partly originate from the choice of the sample group and partly from differences in culture and local habits. 

The present study also collected data regarding some other recognized dementia risk factors [13]. Almost half of the individuals enrolled in our study were smokers (48.5%), an additional alarm sign for the increased dementia risk that individuals visiting outpatient clinics inherently have. Despite vigorous national interventions for smoking cessation, smoking rates are still high, especially in Eastern Europe and Asia. A recent study estimated that approximately 34% of all Romanians over the age of 15 are smokers [35], a result which is comparable to our findings if we adjust for age and study inclusion criteria. In terms of depression, although we did not identify middle-aged individuals with depression, over 40% of the elderly individuals had abnormal scores in the GDS assessment. Although the percentage seems high, it is in line with current reports and meta-analyses that estimate a 40.78% incidence of depression in elderly individuals residing in developing countries [36]. The relationship between dementia and depression is complex and not yet fully understood. While depression causes social withdrawal, changes in mood, and can increase the risk of cognitive dysfunction, most available data exploring the two conditions only offer short-term follow-up [37] and cannot conclusively determine the correlation between these two conditions. Additionally, a large community-based study found no relationship between depressed mood and subsequent dementia [38]. Nonetheless, an abnormal GDS score should act as an alarm sign for the treating physician due to the significant detrimental effect that depression can have on quality of life and general health. Similarly, abnormal MMSE findings should also prompt physicians to recommend additional testing. Even though a prior diagnosis of any degree of cognitive impairment was a predefined exclusion criterion, we still identified 13 individuals with mild or moderate cognitive impairment, equivalent to 7.7% of the entire study population. This finding is in line with the available literature data—a large Cochrane review summary included 15 MMSE screening studies and reported a median prevalence of an abnormal MMSE score of 7.4% [39]. 

Nutritional assessment, although not formally part of any dementia screening tool, identified that over 30% of our study group were malnourished or at risk for malnutrition. Although this finding is somewhat unexpected given the high proportion of obese individuals and the group’s generally good physical health, it is explicable in light of potentially worsened unhealthy eating habits in the post-COVID environment [40]. There is strong evidence favoring the association between decreased dementia risk and a healthy lifestyle that encompasses physical activity, a balanced diet, and social engagement [41]. Most European countries have some type of primary prevention program aimed at encouraging healthy behavior, either through television advertisements, free medical check-ups, specially designed public areas for fitness, etc. Although these are all salutary initiatives, they only appeal to a certain segment of the population. Midlife and elderly individuals are more likely to have more difficulties adhering to long-term lifestyle changes, and an individualized program is often required to address their needs [42]. Since not all countries can provide this for all their citizens, we suggest identifying those most at risk by implementing simple screening tools that can be used for all outpatients visiting a clinic. This strategy has been previously assessed in other populations with good results. Tai and colleagues conducted a screening study in an outpatient department of a regional hospital and assessed consecutive individuals seeking medical help with the aid of the Ascertainment of Dementia 8 (AD8) instrument. The authors found that the ratio of suspected dementia is significantly higher in this pre-selected group than in the general population [19]. Similar results were reported in a geriatric outpatient clinic catering to individuals with memory complaints [43].

Early identification of individuals at risk for dementia has proven its value across multiple settings. A large study that included over 1000 adult individuals with mild cognitive dysfunction assessed participants recruited from several AD centers at baseline and yearly thereafter for three years. The authors reported that after two years, 14% of study participants reverted to normal cognition and 51% did not progress to more severe impairment [44]. Predictors associated with a good outcome included younger age, being unmarried, and higher delayed verbal memory test scores. Another study estimated the impact of addressing modifiable dementia risk factors and concluded that a 5% drop in obesity rates would lower dementia prevalence by 6% and a 5% drop in physical inactivity rate would reduce dementia by 11% [45]. Primary prevention has been confirmed as a very effective strategy for decreasing vascular dementia and AD incidence in several European countries [46]. Currently, Romania does not have a national dementia strategy, despite several initiatives from both patient organizations and professional societies. Additionally, there is a paucity of regional data [47] in terms of the incidence and prevalence of individuals at risk for dementia, which is detrimental to long-term national health policies. Applying the mCAIDE score in daily clinical practice is easy, feasible, and has long-term benefits for midlife and elderly individuals seeking medical care for cardio-vascular or non-cardiovascular conditions. In addition to controlling diabetes, arterial tension, and cholesterol levels, patients identified as having a high dementia risk should receive additional personalized counseling for addressing the other modifiable risk factors, such as weight or physical activity, and should have more rigorous follow-ups for the early detection of dementia symptoms.

This study has several limitations. We do not currently have long-term follow-up data for the patients that were screened for dementia risk. However, they are included in a prospectively-maintained database that is constantly being updated. Additionally, the sample size is quite small, and there is a selection bias inherent to including only patients who voluntarily seek medical help, suggesting that the true number of individuals at risk for developing dementia is significantly higher. Lastly, we did not assess all of the twelve factors currently associated with increased dementia risk due to the initial design of the study protocol. Nonetheless, the present study is among the first to use mCAIDE in a population other than that used for its initial validation, confirming its major potential for wide clinical use. The mCAIDE tool is easy to use and does not take a lot of time. Additionally, because it assesses dyslipidemia based on self-reported values, it does not require drawing a blood sample, which is an advantage in low-resource settings or in outpatient clinics [23]. No statistically significant difference was identified between self-reported cholesterol values and the cholesterol values as assessed by a blood test, suggesting that self-reported cholesterol is a valid alternative for all clinicians. In our study, the mCAIDE results correlated very well with the original CAIDE tool and with other dementia risk factors such as preexisting cognitive dysfunction and diabetes. Using a cut-off value of ≥ 7, physicians can determine which individuals should be referred for additional testing or assessed in a dementia clinic.

## 5. Conclusions

Early identification of modifiable dementia risk factors can significantly decrease dementia incidence in later life. Screening for at-risk individuals in outpatient clinics is an easy-to-implement approach feasible for virtually all healthcare systems. The mCAIDE tool used in the present study can correctly predict dementia, takes little time, and requires no formal biochemical testing. Additional long-term validation is required, as well as longitudinal studies confirming the clinical impact of changing dementia risk factors.

## Figures and Tables

**Table 1 jpm-13-01630-t001:** Demographic and clinical data of the study population.

	Mean (Min/Max)		No. Participants (%)
Age	64.43 ± 9.08 years (40–75 years)	Gender	
		Male	73 (43.2%)
		Female	96 (56.8%)
Systolic blood pressure	146.39 ± 21.9 mm Hg (100–200 mm Hg)	Smoking	
		YES	82 (48.5%)
		NO	87 (51.5%)
Cholesterol	187.85 ± 49.73 mg/dL (86–328 mg/dL)	MMSE	
		Normal	156 (92.3%)
		Cognitive dysfunction	13 (7.7%)
HDL-chol	48.39 ± 12.57 mg/dL (14.5–86.10 mg/dL)	Blood glucose	
		Normal	126 (74.6%)
		Increased	43 (25.4%)
LDL-chol	112.11 ± 44.15 mg/dL (31–257.40 mg/dL)	History of diabetes	
		YES	73 (43.2%)
		NO	96 (56.8%)
Triglycerides	134.10 ± 58.16 mg/dL (46–426 mg/dL)	Uric acid	
		Normal	143 (84.6%)
		Increased	26 (15.4%)
Brachial circumference	22.66 ± 2.45 cm (20–38 cm)		
Calf circumference	31.46 ± 2.15 cm (28–44 cm)		
MNA	23.89 ± 3.64 points (9.5–30 points)		

Chol—cholesterol, MMSE—Mini-Mental State Evaluation, MNA—Mini Nutritional Assessment.

**Table 2 jpm-13-01630-t002:** mCAIDE score results.

	mCAIDE Scoring System	No. Participants (%)
Gender	Male (1 point)	73 (43.2%)
	Female (0 points)	96 (56.8%)
Age	<65 years (0 points)	61 (26.1%)
	65–72 years (1 point)	78 (46.2%)
	≥73 years (2 points)	30 (17.7%)
Education	<12 years (2 points)	110 (64.7%)
	12–16 years (1 point)	38 (22.4%)
	>16 years (0 points)	21 (12.4%)
BMI	≤30 kg/m^2^ (0 points)	66 (39.1%)
	>30 kg/m^2^ (2 points)	103 (60.9%)
Blood Pressure	<140 mm Hg (0 points)	37 (21.9%)
	≥140 mm Hg (2 points)	132 (78.1%)
Cholesterol (self-reported)	No abnormal values (0 points)	72 (42.6%)
	Abnormal values (2 points)	97 (57.4%)
mPPT	≥12 points (0 points)	139 (82.2%)
	<12 points (3 points)	30 (17.8%)

BMI—body mass index.

**Table 3 jpm-13-01630-t003:** The mCAIDE versus CAIDE score in non-elderly individuals.

	mCAIDE Scoring System	Number of Patients (%)	CAIDE Scoring System	Number of Patients (%)
Gender	Male (1 point)	27 (44.3%)	Male (1 point)	27 (44.3%)
	Female (0 points)	34 (55.7%)	Female (0 points)	34 (55.7%)
Age	<65 years (0 points)	61 (100%)	<47 years (0 points)	14 (22.9%)
	65–72 years (1 point)	0 (0%)	47–53 years (3 point)	15 (24.6%)
	≥73 years (2 points)	0 (0%)	>53 years (4 points)	32 (52.5%)
Education	<12 years (2 points)	42 (68.9%)	0–6 years (3 points)	7 (11.5%)
	12–16 years (1 point)	10 (16.4%)	7–9 years (2 point)	35 (57.4%)
	>16 years (0 points)	9 (14.8%)	≥10 years (0 points)	19 (31.1%)
BMI	≤30 kg/m^2^ (0 points)	13 (21.3%)	≤30 kg/m^2^ (0 points)	13 (21.3%)
	>30 kg/m^2^ (2 points)	48 (78.7%)	>30 kg/m^2^ (2 points)	48 (78.7%)
Blood Pressure	<140 mm Hg (0 points)	18 (29.5%)	<140 mm Hg (0 points)	18 (29.5%)
	≥140 mm Hg (2 points)	43 (70.5%)	≥140 mm Hg (2 points)	43 (70.5%)
Cholesterol	Normal (self-reported) (0 points)	39 (63.9%)	≤6.5 mmol/L (0 points)	25 (41.0%)
	Abnormal (self-reported) (2 points)	22 (36.1%)	>6.5 mmol/L (2 points)	36 (59.0%)
Physical activity	≥12 points (0 points)	33 (54.1%)	Active (0 points)	28 (45.9%)
	<12 points (3 points)	28 (45.9%)	Inactive (1 point)	33 (54.1%)
Score	7.48 ± 2.47 points		7.98 ± 2.72 points	

BMI—body mass index.

## Data Availability

The data are not publicly available due to privacy reasons. The data presented in this study are available on request from the corresponding author.

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
