# Peer review of "Preventing Dementia—A Cross-Sectional Study of Outpatients in a Tertiary Internal Medicine Department"

_jpm, 2023, doi:10.3390/jpm13121630_

Round 1

Reviewer 1 Report

Comments and Suggestions for Authors

The revised manuscript was clear. There were some minor typos, for example, Page 3 line 143: Toleal&al.

Please check the manuscript for any typos. Some discussion about relationship between high systolic pressure and dementia can be helpful to add in the introduction.

Some 2-3 sentences about future directions at the end of the manuscript can be helpful.

Author Response

We would like to thank reviewer 1 for the comments. Please find a point-by-point response below:

  1. There were some minor typos, for example, Page 3 line 143: Toleal&al. Please check the manuscript for any typos.

Toleal&al. was changed to Tolea and colleagues. Manuscript was checked for typos.

  1. Some discussion about relationship between high systolic pressure and dementia can be helpful to add in the introduction.

Thank you for your suggestion. We have added a paragraph about the relationship between hypertension and dementia in the Introduction section.

  1. Some 2-3 sentences about future directions at the end of the manuscript can be helpful.

Thank you for the comment. We have added some potential future directions at the end of the manuscript.

Reviewer 2 Report

Comments and Suggestions for Authors

Even though the study is important for dementia prevention and screening in limited-resources countries, the manuscript has some shortages.

In the abstract there is lack of information about the methods.

Study design is not well described and the description of study methods is scarce. There is no explanation on how mCAIDE results are interpreted. What are the level for intermediate or high risk for dementia? It becomes clear only in the results part.

In all the manuscript authors mention elderly and non-elderly population, but it is not well defined why it is necessary. The result part is scattered.

There are no conclusion part?

Comments on the Quality of English Language

Some grammar and logic mistakes are in the manuscript. The thorough editing of English language is strongly recommended.

Author Response

We would like to thank reviewer 2 for the insightful and useful comments of our paper. Below is a point-by-point response to his comments:

  1. In the abstract there is lack of information about the methods.

Thank you for your comment. Indeed, due to constraints pertaining to the number of words, we focused on explaining the aim of the study and the main findings, most likely leaving out data from the material and method section. We have revised the Abstract accordingly, adding more information about study methods.

  1. Study design is not well described and the description of study methods is scarce. There is no explanation on how mCAIDE results are interpreted. What are the level for intermediate or high risk for dementia? It becomes clear only in the results part.

Thank you for your comment. We have revised the Material and Method section, adding information about mCAIDE and CAIDE results and their significance

  1. In all the manuscript authors mention elderly and non-elderly population, but it is not well defined why it is necessary. The result part is scattered.

Thank you for your comment. The distinction between the elderly and the non-elderly population of our study was necessary when comparing CAIDE to mCAIDE because the CAIDE tool is only validated in non-elderly individuals. However, due to population aging, we believe that all individuals, even those aged over 65, should undergo dementia risk screening and thus we decided to use the mCAIDE score in our study. Nonetheless, the mCAIDE has been used in significantly fewer studies, so we felt that comparing it to the traditional CAIDE tool could be useful; however, we could only do the comparison for the non-elderly population of our study, hence the separate headings in the Results section and the scattering of the results. Another elderly versus non-elderly distinction was made in the depression screening – elderly individuals were assessed by means of the GDS and non-elderly were assessed by means of the HADS tool. This distinction is due to hospital policy, which uses different depression questionnaires for elderly and non-elderly individuals.

  1. There are no conclusion part?

Thank you for the comment. We have added a Conclusions section dedicated to summarizing our findings.

  1. Some grammar and logic mistakes are in the manuscript. The thorough editing of English language is strongly recommended.

Thank you very much for your comments. We have thoroughly revised the manuscript for language issues (the changes are tracked in the revised version).

Reviewer 3 Report

Comments and Suggestions for Authors

Dear Editor,

The manuscript by Sandu et al., Preventing dementia – a cross-sectional study of outpatients in a tertiary internal medicine department describes the implementation of a cross-sectional study in an outpatient department for identifying individuals at risk for dementia. The authors present evidence to support the use of a simple questionnaire such as the mCAIDE in a predefined high-risk population because of its ease and low cost. The manuscript is stepwise structured (though some sections need to be looked at again), and the experiments are carefully designed to match the conclusions that are given.

I have carefully read through the manuscript and have these comments to help to improve understanding and add clarity to the work.

1.     Information on the number of morbidity and mortality due to dementia should be included in the Abstract and in the Introduction

2.     In the Abstract, the authors should clarify what is meant by “abnormal results in the Geriatric Depression Scale”

3.     The referencing in the Introduction is really sparse and should be improved on. The same applies to the Discussion.

4.     In the Introduction, the authors should indicate some drugs that have been tested or are currently used to mitigate the effects of dementia.

5.     In lines 68-70, the authors attribute the declines in age-specific dementia to 3 key factors; meanwhile, they earlier mentioned 12 risk factors (60-63). This seems to me to be a disparity, and the authors need to throw more light on this.

6.     The authors should include the full meaning of CAIDE when using it for the first time. Ideally, each time the tool name is given for the first time, it should have the full name and the abbreviation, and then subsequently the abbreviations can be used.

7.     In the Materials and Methods, the authors should clearly state how socio-demographic data was obtained—did they use questionnaires or interviews?

8.     In line 143, they could use, “Tolea and colleagues”. The same approach for line 299

9.     In the Results, I am concerned about whether the authors found any persons who were underweight in their study.

10.  In the Discussion, the authors should give the source of the reports on obesity in Romania (line 247)

11.  In the Discussion, the authors should give details about the large study that reported a reversal of dementia symptoms. This should include, where the study was done, the age population included as well as the intervention(s) (lines 305-308)

12.  Lastly, the paper has grammatical errors in so many places, which sometimes obscure the meaning and make understanding difficult. It will be important for the authors to do proper proofreading.

Comments on the Quality of English Language

12.  Lastly, the paper has grammatical errors in so many places, which sometimes obscure the meaning and make understanding difficult. It will be important for the authors to do proper proofreading.

Author Response

We would like to thank reviewer 3 for the insightful and useful comments of our paper. Below is a point-by-point response to his comments:

  1. Information on the number of morbidity and mortality due to dementia should be included in the Abstract and in the Introduction

Thank you for the comment. We have added the information in both Abstract and Introduction.

  1. In the Abstract, the authors should clarify what is meant by “abnormal results in the Geriatric Depression Scale”

Thank you for the comment. The sentence in the Abstract section has been changed to “Over 40% of the elderly individuals in the study were assessed as “somewhat depressed” or “depressed” on the Geriatric Depression Scale.”

  1. The referencing in the Introduction is really sparse and should be improved on. The same applies to the Discussion.

Thank you for your comment. After revising both sections, taking into account the valuable suggestions from all the reviewers, several references have been added in both the Introduction and Discussion sections of the manuscript.

  1. In the Introduction, the authors should indicate some drugs that have been tested or are currently used to mitigate the effects of dementia.

Thank you for your comment. We have added the required information in the Introduction section.

  1. In lines 68-70, the authors attribute the declines in age-specific dementia to 3 key factors; meanwhile, they earlier mentioned 12 risk factors (60-63). This seems to me to be a disparity, and the authors need to throw more light on this.

Thank you for your comment. The disparity is due to a poor choice of words on our part. While all the twelve risk factors contribute to dementia, diabetes, hypertension and literacy are three risk factors that have significantly changed globally in terms of better disease management and overall increase in education. Some of the other risk factors, such as smoking or obesity are not decreasing in terms of incidence globally, with some regions noting an increase in obesity and a plateau in smoking. Additionally, because some of the risk factors have only recently been identified, there is less literature about them. In order to clarify the issue, we have added the sentence “although changes in other of the afore-mentioned risk factors could also play a role.”

  1. The authors should include the full meaning of CAIDE when using it for the first time. Ideally, each time the tool name is given for the first time, it should have the full name and the abbreviation, and then subsequently the abbreviations can be used.

Thank you very much for your comment. We have included the full meaning of the CAIDE acronym when first used. Additionally, we have checked the manuscript for other abbreviations/acronyms not previously explained.

  1. In the Materials and Methods, the authors should clearly state how socio-demographic data was obtained—did they use questionnaires or interviews?

Thank you very much for your comment. We have added in the Material and Method section a phrase clearly stating that socio-demographic data was obtained by means of interview as per hospital protocol.

  1. In line 143, they could use, “Tolea and colleagues”. The same approach for line 299

Thank you very much for your comment. We have changed the text accordingly.

  1. In the Results, I am concerned about whether the authors found any persons who were underweight in their study.

Thank you very much for the question. Indeed, we did not find any underweight patients, a finding that surprised us as well, especially since there were several individuals at risk for malnutrition. However, we tried to explain our findings in the Discussions section, attributing them to unhealthy eating habits possibly exacerbated by the COVID pandemic.

  1. In the Discussion, the authors should give the source of the reports on obesity in Romania (line 247)

Thank you for your comment. We have revised the manuscript accordingly.                        

  1. In the Discussion, the authors should give details about the large study that reported a reversal of dementia symptoms. This should include, where the study was done, the age population included as well as the intervention(s) (lines 305-308)

Thank you for your comment. We have revised the manuscript accordingly.

  1. Lastly, the paper has grammatical errors in so many places, which sometimes obscure the meaning and make understanding difficult. It will be important for the authors to do proper proofreading.

Thank you very much for your comments. We have thoroughly revised the manuscript for language issues (the revised version has the track changes function enabled).

Round 2

Reviewer 2 Report

Comments and Suggestions for Authors

Congratulations on the big work done.